# cfDNA Methylation Profiles and T-Cell Differentiation in Women with Endometrial Polyps

**DOI:** 10.3390/cells11243989

**Published:** 2022-12-09

**Authors:** Xiao-Hong Li, Mei-Yin Lu, Jia-Li Niu, Dong-Yan Zhu, Bin Liu

**Affiliations:** 1Department of Reproductive Health, Shenzhen Baoan Women’s and Children’s Hospital, Jinan University, Shenzhen 518102, China; 2Department of Biobank, Shenzhen Baoan Women’s and Children’s Hospital, Jinan University, Shenzhen 518102, China

**Keywords:** endometrial polyp, cfDNA, methylation, T-cell differentiation, etiology

## Abstract

DNA methylation is a part of the regulatory mechanisms of gene expression, including chromatin remodeling and the activity of microRNAs, which are involved in the regulation of T-cell differentiation and function. However, the role of cfDNA methylation in T-cell differentiation is entirely unknown. In patients with endometrial polyps (EPs), we have found an imbalance of T-cell differentiation and an aberrant cfDNA methylation profile, respectively. In this study, we investigated the relationship between cfDNA methylation profiles and T-cell differentiation in 14 people with EPs and 27 healthy controls. We found that several differentially methylated genes (DMGs) were associated with T-cell differentiation in people with EPs (*ITGA2*-Naïve CD4, *r =* −0.560, *p =* 0.037; *CST9*-EMRA CD4, *r =* −0.626, *p =* 0.017; and *ZIM2*-CM CD8, *r =* 0.576, *p =* 0.031), but not in healthy controls (all *p* > 0.05). When we combined the patients’ characteristics, we found a significant association between *ITGA2* methylation and polyp diameter (*r =* 0.562, *p =* 0.036), but this effect was lost when adjusting the level of Naïve CD4 T-cells (*r =* 0.038, *p =* 0.903). Moreover, the circulating sex hormone levels were associated with T-cell differentiation (estradiol-Naïve CD4, *r =* −0.589, *p =* 0.027), and the cfDNA methylation profile (testosterone-*ZIM2*, *r =* −0.656, *p =* 0.011). In conclusion, this study has established a link between cfDNA methylation profiles and T-cell differentiation among people with EPs, which may contribute to the etiology of EPs. Further functional studies are warranted.

## 1. Introduction

An endometrial polyp (EP) is a common gynecological disease, which refers to the overgrowth of endometrial glands and stroma, protruding into the uterine cavity [1]. EPs seriously affect women’s reproductive health, which can lead to abnormal uterine bleeding (AUB), implantation failure, abortion, other symptoms, and even cancer [2,3]. However, the pathogenesis of EPs is still unclear. However, some factors, such as aging, obesity, hypertension, and usage of drugs are considered risk factors for EPs [4]. Importantly, the imbalance of immune function may also be an influential factor leading to EPs [5]. In our previously published study, we found an imbalance of T-cell differentiation and function in patients with EPs [6].

As an essential and stable epigenetic mechanism, DNA methylation is mediated by DNA methyltransferases (DNMTs) and is a pivotal regulatory mechanism of gene expression similar to chromatin remodeling and microRNAs, and is crucial to regulating cell development [7,8] and immune function [9,10]. It has been reported to be involved in the differentiation and functional regulation of T-cells [11,12,13].

Cell-free DNA (cfDNA) is a short DNA fragment from apoptotic cells and exists in plasma [14]. Because DNA methylation has high tissue specificity, cfDNA methylation has become a biomarker widely used in the clinical detection of various diseases [8,15]. We have found an aberrant methylation profile of cfDNA in 45 people with EPs, compared with 55 healthy female controls (submitted). However, the link between cfDNA methylation and T-cell differentiation is still unknown.

In this study, we investigated the relationship between the profiles of cfDNA methylation and T-cell differentiation/function in 14 patients with EPs and 27 healthy controls and analyzed the correlation between patients’ clinical characteristics and the abovementioned indices of methylation and T-cells. Herein we report our first study on cfDNA methylation and T-cell differentiation in EPs.

## 2. Materials and Methods

### 2.1. Participants

Fourteen incident people with EPs aged 25∼36 years old from Shenzhen Baoan Women’s and Children’s Hospital, Jinan University, were enrolled in this study between July 2019 and October 2019. All patients were diagnosed by hysteroscopy, ultrasound imaging, and pathology. After signing a written informed consent form, a 5 mL anticoagulant blood sample was collected from each participant before any treatment to assess the phenotype of peripheral immune cells and the cfDNA methylation profile. Demographic data collected from the medical records of the patients included age, menstrual cycle data, reproductive history, blood tests, and clinical characteristics of EPs.

Meanwhile, 27 healthy females of reproductive age who were about to receive pre-pregnancy medical examinations were also randomly enrolled in the study as a control group. Exclusion criteria included EPs, uterine diseases, ovarian diseases, and other infertility disorders. Similarly, a 5 mL anticoagulant blood sample was collected for immunocyte flow cytometry and DNA methylation analysis. Additionally, the age, menstruation, reproductive history, and other information of the control group were obtained through a questionnaire.

This study was approved by the Ethics Committee of Shenzhen Baoan Women’s and Children’s Hospital, Jinan University (IRB No: LLSC-2018-08-01).

### 2.2. Flow Cytometry Analysis

We used the mononuclear cells isolated from each subject to measure T-cell differentiation and function in patients with EPs and in the healthy controls using flow cytometry. The cells were stained by the following antibodies: PerCP-Cy5.5-conjugated anti-CD3; APC-conjugated anti-CD4; BV510-conjugated anti-CD8; FITC-conjugated anti-CD45RA; Alexa Fluor 647-conjugated anti-CCR7 and anti-CXCR5; BB515-conjugated anti-PD-1; PE-conjugated anti-CD25; PE-CY7-conjugated anti-CD28; and V421-conjugated anti-CD127. These were purchased from BD Biosciences. Based on our previously published research [6,16], all of the samples were tested on a BD LSR Fortessa cell analyzer (BD Biosciences, Franklin Lakes, NJ, USA) at Shuangzhi Purui Medical Laboratory Co., Ltd. (Wuhan, China), and data were analyzed using FlowJo 10.1 software (Tree Star Inc., Ashland, OR, USA).

### 2.3. cfDNA Methylation Analysis

cfDNA was extracted from plasma using the MagaBio plasma circulating DNA purification kit (Bioer Tech., Hangzhou, China). The genome-wide methylation analysis of cfDNA was performed by MethylGene Tech Co. Ltd. (Guangzhou, China) by targeted capture methylation sequencing [17]. Briefly, 5 ng cfDNA was treated by bisulfite and used for methylation library construction using MethylGene Ultralow EM-Seq Library Prep Kit (MethylGene Tech Co., Ltd.). We analyzed the constructed libraries using an Agilent 2100 Bioanalyzer (Agilent Tech., PaloAlto, CA, USA) and performed liquid-hybridization-based capture of methylation libraries using SeqCap EZ Hybridization and Wash Kits (Roche NimbleGen, Roche Ltd., Basel, Switzerland). The sequencing of these libraries was performed on an Illumina Nova-Seq 6000 (Illumina, San Diego, CA, USA) using the PE150 model. The sequencing reads were filtered to exclude the low-quality reads (Phred score < 5), and then were aligned to the human reference genome (hg19) using BSMAP (Version 2.74) [18]. We identified the differentially methylated CpGs (DMCs) using metilene (Version 0.2-6) [19], selected the CpGs that were covered by more than 10× sequence reads, and reached the thresholds of differential methylation β ≥ 15% between the patients and controls for gene annotation. We then selected genes with more than one DMC and at least one DMC with a methylation difference > 0.5 for further analyses. The false discovery rate (FDR) was calculated using the Benjamini–Hochberg procedure [20], and only CpG sites with *p* < 0.05 were selected for further analyses.

Validation of DMCs was performed using multiplex amplicon methylation PCR sequencing on all subjects. Two rounds of PCR were performed on enzymatically converted cfDNA by NEBNext^®^ Enzymatic Methyl-seq Kit (EM-Seq, NEB). The multiplex amplicon PCR used the primers listed in Appendix A, which were ligated with the adaptor sequence for the Illumina platform at their 5′ ends in silicon (Forward: 5′-CCTAC ACGAC GCTCT TCCGA TCT-3′; Reverse: 5′-TTCAG ACGTG TGCTC TTCCG ATCT-3′). After purification, the PCR products were ligated to TruSeq Dual Index Adaptors (Illumina, San Diego, CA, USA) by the second-round PCR using Illumina index primer (5′-CAAGC AGAAG ACGGC ATACG AGAT-index-GTGAC TGGAG TTCAG ACGTGT GCTCT TCCGA TCT-3′ and 5′-AATGA TACGG CGACC ACCGA GATCT ACAC-index-ACACT CTTTC CCTAC ACGAC GCTCT TCCGA TCT-3′). Finally, the purified libraries were sequenced by Illumina Nova-Seq 6000 using the PE150 model.

Further, GO and KEGG enrichment of validated DMCs was analyzed using WebGestalt (WEB-based Gene SeT AnaLysis Toolkit) [21].

### 2.4. Statistical Analysis

SPSS 27.0 (IBM Corp., Armonk, NY, USA) was used for data processing and analysis. The Mann–Whitney U test was performed to analyze the differences in cfDNA methylation, immune cells, and their subsets between the patients and controls. Furthermore, the relationship between the above indices and clinical phenotypes was analyzed using Spearman correlation analyses. All tests were two-sided, with the level of significance set at 0.05. Statistical figures were produced by GraphPad Prism9.4 (GraphPad Software, San Diego, CA, USA). 

## 3. Results

### 3.1. Characteristics of the Study Populations

Table 1 summarizes the clinical and demographic characteristics of 14 people with EPs and 27 healthy controls. In this study, the menstrual duration of people with EPs was shorter than that of the healthy controls (*p =* 0.042), but it was within the normal range (4–7 days). However, there was no significant difference in the distribution of other listed demographic characteristics between the case group and the control group (all *p* values > 0.05).

### 3.2. Analysis of T-Cell Immune Function in Patients with EPs and Controls

According to our previous method [6], we used flow cytometry to analyze Naïve CD4+ T-cells (CD4+CCR7+CD45RA+), central memory (CM) CD4+ T-cells (CD4+CCR7+CD45RA−), terminally differentiated effector memory (EMRA) CD4+ T-cells (CD4+CCR7−CD45RA+), CM CD8+ T-cells (CD8+CCR7+CD45RA−), EMRA CD8+ T-cells (CD8+CCR7−CD45RA+), double-negative (DN) T-cells (CD3+CD4−CD8−), Th1/Th2 ratio, Th1 + Th17/Th2 ratio, and Tfh1/Tfh2 ratio between 14 people with EPs and 27 control groups. As shown in Figure 1, we found that ratios of Naïve CD4+ T-cells, CM CD4+ T-cells, and CM CD8+ T-cells were significantly higher in the people with EPs than in the healthy controls. As for the ratio of EMRA CD4+ T-cells, it was significantly decreased in the patients. However, there was no statistical difference in EMRA CD8+ T-cells, the Th1/Th2 ratio, the Th1 + Th17/Th2 ratio, and the Tfh1/Tfh2 ratio between the people with EPs and the control patients (Appendix A).

### 3.3. Differences in cfDNA Methylation between Patients with EP and Controls

In our previous study, we found that 19 genes (*IGF1R*, *CTBP1*, *TCF7L1*, *E2F3*, *CACNA2D4*, *KCNJ12*, *TPO*, *UGT1A8/10*, *CABP5*, *CST9*, *ITGA2*, *DLGAP2*, *ESPNP*, *NBPF25P*, *RASA3*, *ZIM2*, *PXDN*, *HDAC4*, and *VAV2*) were hypomethylated in 45 EP cases, compared with 55 female controls (submitted). Basic information on these DMGs and related DMCs was described in Appendix A. As shown in Figure 2, the Mann–Whitney U tests indicated that the methylation levels of these 19 genes in 14 patients were lower than those in the 27 controls. Furthermore, we found that these DMGs were partly related to the ITGA2 pathway (GO:0005886 and GO:0016021, both *p* values < 10^−11^).

Moreover, 11 of the 19 genes above (*IGF1R*, *CTBP1*, *TCF7L1*, *E2F3*, *ITGA2*, *HDAC4*, *TPO*, *CABP5*, *ZIM2*, *PXDN*, and *VAV2*) are found in the methylation data of the GDC TCGA Endometrioid Cancer cohort (UCEC, n = 606) using UCSC Xena software (https://xenabrowser.net/, accessed on 28 November 2022). As shown in Appendix A, the methylation levels of 7 genes (*IGF1R*, *CTBP1*, *TCF7L1*, *E2F3*, *ITGA2*, *HDAC4*, and *VAV2*) in the UCEC data are extremely low (β < 0.1). Of the other 4 genes, the methylation levels of 3 (*TPO*, *ZIM2*, and *PXDN*) in the UCEC data are significantly lower than the 27 healthy controls in this study (all *p* values < 0.05, tested by Student *t* tests). In total, 10 of 19 DMGs are hypomethylated in UCEC. Meanwhile, the mRNA levels of 8 genes (*IGF1R*, *CTBP1*, *TCF7L1*, *E2F3*, *ITGA2*, *HDAC4*, *PXDN*, and *VAV2*) are highly expressed in UCEC (all log2(counts) values > 8).

### 3.4. Correlation Analysis between cfDNA Methylation and T-Cell Differentiation

Subsequently, based on the above results, we analyzed the correlations between cfDNA methylation and T-cell differentiation using the Spearman correlation. As shown in Figure 3, the methylation level of the *ITGA2* gene was negatively correlated with the ratio of Naïve T CD4+ cells (Spearman *r* = −0.560, *p* = 0.037). Additionally, a lower methylation level of the *CST9* gene was observed in the patients with higher levels of EMRA CD4+ T-cells (Spearman *r* = −0.626, *p* =  0.017). Moreover, we found that the methylation level of the *ZIM2* gene was positively correlated with the ratio of CM CD8+ T-cells (Spearman *r* = 0.576, *p* = 0.031). In the control group, however, we did not find any significant correlation (all *p* values > 0.05).

### 3.5. Correlation Analysis of Clinical Indicators of EP with cfDNA Methylation and T-Cell Differentiation

We further analyzed the correlations of clinical indicators with cfDNA methylation and T-cell differentiation in the patients (Table 2). As shown in Table 2a, we noticed that the hormone levels were negatively correlated with DNA methylation levels and T-cell subsets in the patients, respectively. The circulating level of estradiol (E2) was negatively associated with the ratio of Naïve CD4+ T-cells (Spearman *r* = −0.589, *p* = 0.027). The serum level of testosterone was negatively associated with the methylation level of the *ZIM2* gene (Spearman *r* = −0.656, *p* = 0.011). For other clinical indicators, we observed that the level of *ITGA2* methylation was positively correlated with the diameter of the EPs, as shown in Table 2b (Spearman *r* = 0.562, *p* = 0.036). After adjusting the Naïve CD4+ T-cells, the partial correlation coefficient between the EP’s diameters and ITGA2 was not significant (*r =* 0.038, *p =* 0.903).

## 4. Discussion

In our previous study and other published studies, EP was associated with an imbalance of immune cells [6,22,23], which can be regulated by abnormal DNA methylation [24]. To our knowledge, there are still no reports regarding the association between cfDNA methylation and immune cell differentiation/function. In this study, we combined cfDNA methylation profiles with immune cells, specifically T-cell differentiation to analyze EP. We found that the methylation levels of the *ITGA2*, *CST9*, and *ZIM2* genes in cfDNA had been reduced in people with EPs, and these DMGs were associated with the ratios of Naïve CD4+ T-cells, CM CD4+ T-cells, and CM CD8+ T-cells in 14 patients with EP, respectively, but not in the 27 healthy controls. Moreover, the abovementioned DMGs and T-cell subpopulations were associated with the circulating levels of E2 and testosterone, suggesting that the link between cfDNA methylation and T-cell differentiation may be involved with the etiology of EPs. Additionally, the methylation level of the *ITGA2* gene was associated with polyp size, which interacted with the ratio of Naïve CD4+ T-cells, implying that the role of cfDNA methylation in EP development may be affected by T-cell differentiation.

Firstly, this study showed the imbalance of Naïve CD4+ T-cells, CM CD4+ T-cells, CM CD8+ T-cells, and EMRA CD4+ T-cells in 14 people with EPs compared to 27 healthy controls, which is similar to our previous study [6]. In a published report on endometrial carcinoma [25], Chang et al. found that the circulating CD4+ T-cells expressed significantly higher CCR7+ CD45RO- (a marker for Naïve CD4+ T-cells) and CCR7+ CD45RO+ (a marker for CM CD4+ T-cells), but lower CCR7-CD45RO+ (a marker for EMRA CD4+ T-cells) than the infiltrating CD4+ T-cells; while the circulating CD8+ T-cells expressed higher CCR7+ (markers of the progenitor cells in CD8+ T-cells) than the infiltrating CD8+ T-cells. It seems that the dysdifferentiation of circulating T progenitor cells into EMRA T-cells helps tumor cell proliferation [26]. Combing these reports with our study, we speculated that a higher distribution of Naïve CD4+ T-cells, CM CD4+ T-cells, and CM CD8+ T-cells, and a lower distribution of EMRA CD4+ T-cells in circulating blood are associated with the proliferation of endometrial cells.

Secondly, our study also found that 19 genes were hypomethylated in cfDNA in 14 people with EPs compared with the 27 healthy controls. Among these 19 genes, 10 genes (*IGF1R*, *CTBP1*, *TCF7L1*, *E2F3*, *ITGA2*, *HDAC4*, *TPO*, *ZIM2*, *PXDN*, and *VAV2*) were also hypomethylated in the GDC TCAC UCEC cohort, and 8 genes (*IGF1R*, *CTBP1*, *TCF7L1*, *E2F3*, *ITGA2*, *HDAC4*, *PXDN*, and *VAV2*) were highly expressed in these UCEC patients (Appendix A). This suggested that the DMGs found in this study are at least partly associated with the proliferation of endometrial cells.

Moreover, we found that the methylation levels of 3 genes (*ITGA2*, *CST9*, and *ZIM2*) in cfDNA are associated with the abovementioned imbalance of T-cell differentiation in this study. Integrin alpha 2 (ITGA2, CD49b) is the alpha subunit of a transmembrane receptor that mediates signal transduction within the cell–cell and cell–extracellular matrix (ECM) interactions, which play roles in various cell functions [27,28]. As a cell signal transduction receptor [29,30], ITGA2 forms T-cell immune synapses to activate T-cells and regulate their proliferation, cytokine secretion [31], and apoptosis [32]. This is related to the strong immunosuppressive activity in CD4+1-type T regulatory (Tr1) cells [33]. Endometrial tissue contains various integrins, which are critical to the activation of endometrial T-cells, and thus the pathogenesis of EP [34]. Therefore, our results on the enrichment of the ITGA2 pathway in DMGs in EP are plausible. Moreover, we found that *ITGA2* methylation was associated with Naïve CD4+ T-cells in patients with EP, which was related to the circulating E2 level. Because of high exposure to E2 in EP [35], we hypothesized that an elevated E2 level causes aberrant gene methylation, and hence the imbalance of T-cell differentiation. Additionally, we found that the association between the methylation level of the *ITGA2* gene and polyp size was affected by the role of Naïve CD4+ T-cells. Our results suggest that the tight link between hormone exposure, gene methylation, and immune function plays a pivotal role in the etiology of EP. Because polyp size is an important index of EP recurrence [36], our results might also have implications for disease prognosis.

*ZIM2* is an imprinted gene located 25 kb downstream of a testis-expressing gene *PEG3* (paternally expression gene 3) [37], suggesting its potential function in the testis [38]. However, there have been very few reports of the *ZIM2* gene in human disease or immune function. The GeneCards database shows a direct interaction between ZIM2 and serine/threonine kinase STK3 (https://www.genecards.org/cgi-bin/carddisp.pl?gene=ZIM2#pathways_interactions, accessed on 21 October 2022). STK3 is a key molecule in the Hippo pathway, which is highly expressed in the endometrium [39]. The activity of the STK3/ Hippo pathway is activated through estrogen and its receptor [40,41], and then regulates angiogenesis, cell proliferation and differentiation, and apoptosis in the endometrium, thus increasing the risk of EP. Moreover, the STK3/Hippo pathway participates in immune regulation [42]. Through integrin-mediated adhesion, changes in cell polarity, and the activation of transcription factors, STK3 has the functions of lymphocyte transport, effector differentiation and function, and immune homeostasis and tolerance [43]. It can coordinate the homeostasis of CD8+ T-cells [44], while its overexpression can promote the migration and infiltration of CD8+T-cells in the tumor microenvironment [45]. In our study, the circulating testosterone level was associated with the methylation of the *ZIM2* gene in people with EPs, thus affecting the ratio of CM CD8+ T-cells. We speculate that this role is through the STK3/Hippo pathway.

Cystatin 9 (CST9), an inhibitor of lysosomal cysteine proteases, exists in all body chambers and fluids containing matrix metalloproteinases (MMP). The latter is a cysteine protease secreted by the immune, epithelial, and endothelial cells, that decomposes ECM. CST9 has immune-modulatory effects such as regulating cytokine secretion and immune cell migration in inflammation and infection status [46]. EMRA CD4+ T-cells are important memory immune cells in inflammation and infection [47]. Therefore, our results on the association between the methylation level of the *CST9* gene and the ratio of EMRA CD4+ T-cells in people with EPs are plausible.

However, our study contained some limitations. (1) The sample size of this study was small, and large-scale studies of people with EPs would be warranted in the future to verify the above results. (2) The age distribution of patients with EP is not exactly consistent with that of healthy controls. However, we did not find any significant correlation between age and methylation/immune indices in this study (all *p* values > 0.05). Moreover, the significant associations between the cfDNA methylation profile and T-cell subsets were seen only in the people with EPs, not in the controls. The confounding role of age on the association between the above indexes might be inessential. (3) The scope of our study was limited to several imbalanced T-cell subsets in EP, which was found in our previous study. In the next step, we would need to further study and analyze a broader differentiation and function of T-cells.

## 5. Conclusions

This study was the first to show that the association between cfDNA methylation and T-cell differentiation is crucial to the etiology of EP, which can be affected by sex hormone exposure. The detection of cfDNA methylation and immune cells in peripheral blood, which is a non-invasive method, will be a simple method for disease prediction and assessment.

## Figures and Tables

**Figure 1 cells-11-03989-f001:**
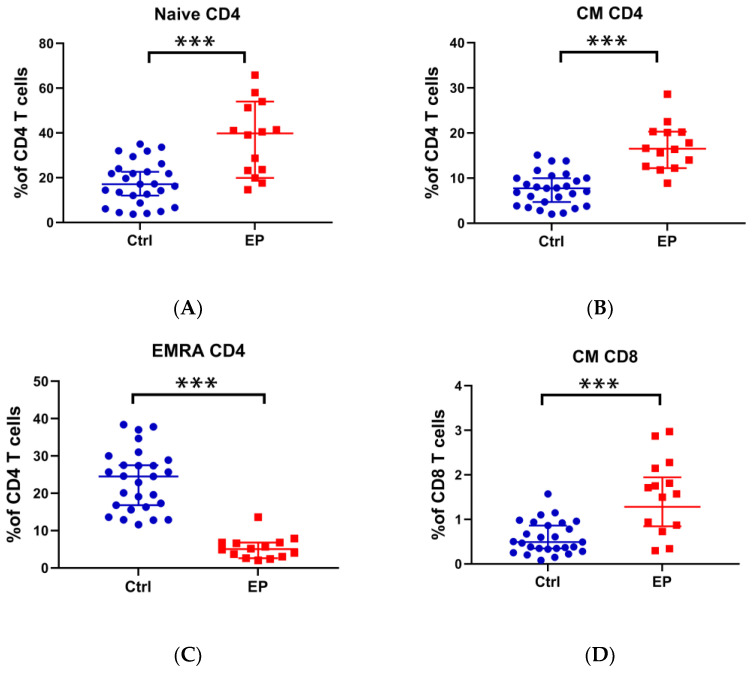
Comparison of the ratio of Naïve CD4+ T-cells, CM CD4+ T-cells, EMRA CD4+ T-cells, and CM CD8+ T-cells between people with EPs and healthy controls. (**A**) Percentage of Naïve CD4+ T-cells in CD4+ T-cells; (**B**) percentage of CM CD4+ T-cells in CD4+ T-cells; (**C**) percentage of EMRA CD4+ T-cells in CD4+ T-cells; (**D**) percentage of CM CD8+ T-cells in CD8+ T-cells. *** *p <* 0.001.

**Figure 2 cells-11-03989-f002:**
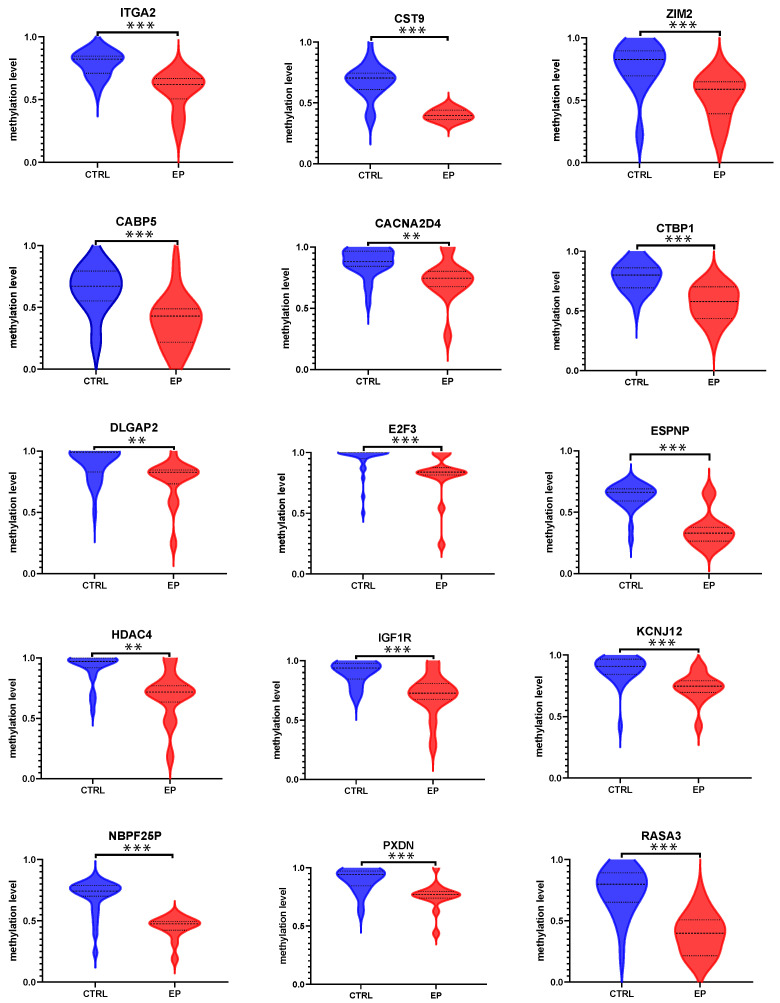
Comparison of methylation levels of *ITGA2*, *CST9*, *ZIM2*, *CABP5*, *CACNA2D4*, *CTBP1*, *DLGAP2*, *E2F3*, *ESPNP*, *HDAC4*, *IGF1R*, *KCNJ12*, *NBPF25P*, *PXDN*, *RASA3*, *TCF7L1*, *TPO*, *UGT1A8/10*, and *VAV2* between people with EPs and healthy controls. ** *p <* 0.01, *** *p <* 0.001.

**Figure 3 cells-11-03989-f003:**
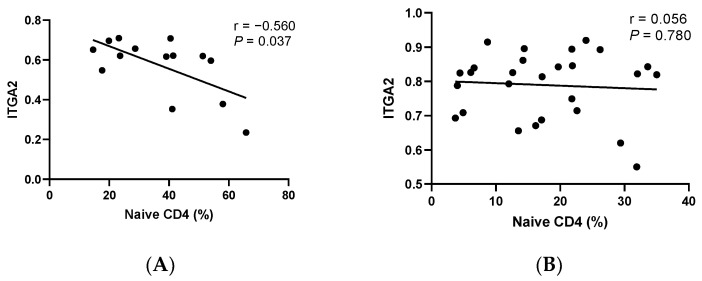
Correlation between cfDNA methylation and T-cell differentiation in people with EPs and controls. (**A**) The relationship between the methylation level of *ITGA2* and the ratio of Naive CD4+ T-cells in people with EPs. (**B**) The relationship between the methylation level of *ITGA2* and the ratio of Naive CD4+ T-cells in healthy controls. (**C**) The relationship between the methylation level of *CST9* and the ratio of EMRA CD4+ T-cells in people with EPs. (**D**) The relationship between the methylation level of *CST9* and the ratio of EMRA CD4+ T-cells in healthy controls. (**E**) The relationship between the methylation level of *ZIM2* and the ratio of CM CD8+ T-cells in people with EPs. (**F**) The relationship between the methylation level of *ZIM2* and the ratio of CM CD8+ T-cells in healthy controls.

**Table 1 cells-11-03989-t001:** Clinical and demographic characteristics of endometrial polyp patients and healthy controls.

Variables	Controls, n (%)	Cases, n (%)	*p* ^a^
All subjects	27(100.0)	14(100.0)	
Age, years			0.269
<30	20(74.1)	7(50.0)	
≥30	7(25.9)	7(50.0)	
Menstrual cycle, days			0.714
24–35	27(100.0)	12(85.7)	
>35	0(0.0)	2(14.3)	
Menstrual duration, days			0.042
≤7	27(100.0)	14(100.0)	
>7	0(0.0)	0(0.0)	
Without dysmenorrhea	26(96.3)	12(85.7)	0.548
AUB	-	2(14.3)	
Gravidities			0.674
0	15(55.6)	8(57.1)	
1	6(22.2)	5(35.8)	
≥2	6(22.2)	1(7.1)	
Abortion			0.471
0	18(66.7)	11(78.6)	
1	6(22.2)	3(21.4)	
≥2	3(11.1)	0(11.0)	
Deliveries			0.968
0	19(70.4)	10(71.4)	
1	8(29.6)	4(28.6)	
Red blood cells (×10^12^/L)			0.714
<4.57	13(48.1)	9(64.3)	
≥4.57	14(51.9)	5(35.7)	
Hemoglobin (g/L)			0.115
<136	13(48.1)	9(64.3)	
≥136	14(51.9)	5(35.7)	
White blood cells (×10^9^/L)			0.796
<6.15	15(55.6)	7(50.0)	
≥6.15	12(44.4)	7(50.0)	
FSH (IU/L)			
<8.01	-	10(71.4)	
≥8.01	-	4(28.6)	
LH (IU/L)			
<11.11	-	11(78.6)	
≥11.11	-	3(21.4)	
Testosterone (nmol/L)			
<1.54	-	10(71.4)	
≥1.54	-	10(71.4)	
E2 (pmol/L)			
<582	-	11(78.6)	
≥582	-	3(21.4)	
Progesterone (nmol/L)			
<8.50	-	13(92.9)	
≥8.50	-	1(7.1)	
Prolactin (μg/L)			
<11.88	-	9(64.3)	
≥11.88	-	5(35.7)	
Endometrial thickness, cm		0.9 ± 0.3	
<0.9	-	6(42.9)	
≥0.9	-	8(57.1)	
Single polyp	-	10(71.4)	
Multiple polyps	-	4(28.6)	
Diameter of polyp, cm	-	1.1 ± 0.6	
>1	-	6(42.9)	
≤1	-	8(57.1)	

Notes: ^a^ Mann–Whitney U-test or Chi-square test. FSH, follicle-stimulating hormone. LH, luteinizing hormone.

**Table 2 cells-11-03989-t002:** Correlation analysis of clinical indexes with T-cell subsets and DNA methylation level.

a. Correlation analysis of Hormone level with T-cell subsets and DNA methylation level				
	*r* ^a^		*p* ^a^	
E2				
Naïve CD4	−0.589		0.027	
Testosterone				
*ZIM2*	−0.656		0.011	
**b. Correlation analysis between methylation level and clinical indexes**				
	***r* ^a^**	***p* ^a^**	***r* ^b^**	***p* ^b^**
*ITGA2*				
Diameter of EP	0.562	0.036	0.038	0.903

Notes: ^a^ Spearman test. ^b^ Partial correlations analysis by controlling Naïve CD4 T-cells.

## Data Availability

The data are not publicly available due to Chinese policy. The raw data supporting the conclusions of this article will be made available by the authors, without undue reservation.

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
