# Peer review of "cfDNA Methylation Profiles and T-Cell Differentiation in Women with Endometrial Polyps"

_cells, 2022, doi:10.3390/cells11243989_

Round 1
Reviewer 1 Report
Presented work describes the results of epigenetic case-control study on the associations between DNA methylation of cfDNA and the content of specialised immune cells in women suffering from Endometrial polyp.
This data are a kind of preliminary results which however are much interesting to be published in the present form.
Please find the comments below:
Line 11: “DNA methylation regulates T cell differentiation and function.”
This chemical modification of DNA is rather a part of epigenetic mechanisms responsible for gene expression. I suggest to correct the expression, like being a part of regulatory mechanisms of gene expression including chromatin remodelling and the activity of microRNAs.
Line 38: “and plays a key role in regulating gene expression, 38 cell development [7,8], and is crucial to regulate immune function [9,10].” I suggest to make the correction similar as comments to line 11
Lines 42,43: Because of its high specificity, cfDNA methylation has become a biomarker 42 widely used in the clinical detection of various diseases [8,15]. “Because of its high specificity” – pleas extend this expression
Author Response
Comments and Suggestions for Authors
Presented work describes the results of epigenetic case-control study on the associations between DNA methylation of cfDNA and the content of specialised immune cells in women suffering from Endometrial polyp.
This data are a kind of preliminary results which however are much interesting to be published in the present form.
Response: We appreciate you for your positive comments.
Please find the comments below:
Line 11: “DNA methylation regulates T cell differentiation and function.”
This chemical modification of DNA is rather a part of epigenetic mechanisms responsible for gene expression. I suggest to correct the expression, like being a part of regulatory mechanisms of gene expression including chromatin remodelling and the activity of microRNAs.
Response: Thank you very much for your kindly suggestions. We have changed the expression like this, “DNA methylation is a part of the regulatory mechanisms of gene expression including chromatin remodeling and the activity of microRNAs, which involve in the regulation of T cell differentiation and function”.
Line 38: “and plays a key role in regulating gene expression, 38 cell development [7,8], and is crucial to regulate immune function [9,10].” I suggest to make the correction similar as comments to line 11
Response: We appreciate you for your good comments and suggestions. We have changed the expression like this, “is a pivotal regulatory mechanism of gene expression similar to chromatin remodeling and microRNAs, and is crucial to regulating cell development [7,8], and immune function [9,10].
Lines 42,43: Because of its high specificity, cfDNA methylation has become a biomarker 42 widely used in the clinical detection of various diseases [8,15]. “Because of its high specificity” – pleas extend this expression
Response: We appreciate you for your suggestions. We are very sorry for our negligence of unclear information. We have changed the expression like this, “Because DNA methylation has high tissue specificity”.
Reviewer 2 Report
In the manuscript, authors were trying to find the association between aberrant methylation profile of cfDNA with the functional T cell subsets in endometrial polyps patients. Authors declared that they found 19 genes hypomethylated in 45 EP patients compared with 55 female normal people (detailed information not shown). Then authors demonstrated that the methylation level of 19 genes could be validated by 14 EP patients and 27 healthy controls and were associated with T cell differentiation in EP patients. Authors declared that they firstly established a link between cfDNA methylation profile and T cell differentiation in EP patients, which may contribute to the etiology of EP. However, 14 EP patients and 27 healthy controls were too few to support the conclusion. More samples are needed to make results credible.
Major concerns:
1. 14 EP patients and 27 healthy control samples were too few to support any of the conclusion in the manuscript. Please expand the sample size and validate your conclusions by other published data sets.
2. The age distribution of patients with EP is not consistent with that of normal controls.
3. How did you find 19 genes that hypomethylated in EP patients? What is the selection criterion? Have you calculated the fdr? How to exclude the false positive genes from all targets?
4. What is the exact hypomethylated position in each gene? Promoter or gene body? Could you please explain the effects of methylation of these genes on the pathogenesis of EP.
5. As the results on the 19 hypomethylated genes in EP patients were important for this manuscript, you need to explain them in detail. Or you could publish the paper about the 19 genes first, and then cite your previous work.
Minor concerns:
1. In part 2.3, please give more details on the validation of DMCs. Did you validate 19 genes in plasma from EP patients and healthy controls by the primers you mentioned in part 2.3?
Author Response
Comments and Suggestions for Authors
In the manuscript, authors were trying to find the association between aberrant methylation profile of cfDNA with the functional T cell subsets in endometrial polyps patients. Authors declared that they found 19 genes hypomethylated in 45 EP patients compared with 55 female normal people (detailed information not shown). Then authors demonstrated that the methylation level of 19 genes could be validated by 14 EP patients and 27 healthy controls and were associated with T cell differentiation in EP patients. Authors declared that they firstly established a link between cfDNA methylation profile and T cell differentiation in EP patients, which may contribute to the etiology of EP. However, 14 EP patients and 27 healthy controls were too few to support the conclusion. More samples are needed to make results credible.
Response: We appreciate you for your kindly comments.
Major concerns:
- 14 EP patients and 27 healthy control samples were too few to support any of the conclusion in the manuscript. Please expand the sample size and validate your conclusions by other published data sets.
Response: We appreciate you for your comments and suggestions. This is true that this study had a small sample size. For our studies on EP, the samples only with cfDNA methylation profile and those only with immune indexes are larger, however samples with both types of data are minute. Recently, we have acquired a new fund on this topic, and will to collect new samples for future study. However, we cannot obtain the new data very quickly, so that this problem might not be solved in this revision. We have discussed this limitation in the Discussion.
- The age distribution of patients with EP is not consistent with that of normal controls.
Response: Thank you very much for your comments.
This is true that the age distribution of patients with EP is not exactly consistent with that of healthy controls in this study. However, we did not find any significant correlation between age and methylation/immune indices in this study (all P values >0.05). Moreover, the significant associations between the cfDNA methylation profile and T cell subsets were seen only in the EP patients, not in the controls. The confounding role of age on the association between above indexes might be inessential.
We have added this limitation in the Discussion part.
- How did you find 19 genes that hypomethylated in EP patients? What is the selection criterion? Have you calculated the fdr? How to exclude the false positive genes from all targets?
Response: We appreciate you for your comments and suggestions. We are sorry for above unclear information.
The genome-wide methylation analysis of cfDNA was performed by MethylGene Tech Co. Ltd. (Guangzhou, China) according to its routine method (ref [17]). Briefly, after excluding low quality reads (Phred score < 5) and aligning to hg19, DMCs were identified using metilene (Version 0.2-6) for CpGs covered by more than 10× sequence reads, applying the thresholds of differential methylation β≥15% between the patients and controls, and adjusted P-value for Mann-Whitney-U-Test <0.05. In this study, only genes with more than one DMC and at least one DMC with a methylation difference > 0.5 were selected for further analyses.
The false discovery rate (FDR) was calculated using the Benjamini & Hochberg procedure, described as [Asomaning N, Archer KJ. High-throughput DNA methylation datasets for evaluating false discovery rate methodologies. Comput Stat Data Anal. 2012 Jun 1;56(6):1748-1756. doi: 10.1016/j.csda.2011.10.020]. Only CpG sites with P<0.05 were selected for further analyses.
We have added the above information in the part of Methods.
- What is the exact hypomethylated position in each gene? Promoter or gene body? Could you please explain the effects of methylation of these genes on the pathogenesis of EP.
Response: Thank you very much for your comments.
According to your suggestions, we have added a Supplementary Table to state the exact hypomethylated position in each gene, whether in the promoter or genebody.
We have explained the possible roles of these 19 DMGs in the etiology of EP in another submitted manuscript regarding the etiology and biomarkers of EP. In fact, these 19 DMGs are majorly enriched in the IGF1R- and ITGA2- pathways, according to the GO and KEGG analyses. Both of these two pathways are related to cell proliferation, thus the risk of EP. The genes related to these two pathways has been emphasized in the submitted manuscript.
In this manuscript, three hypomethylated genes (ITGA2, CST9, and ZIM2) were associated with T-cell differentiation and function. Therefore, we have mainly discussed these three genes in this manuscript.
- As the results on the 19 hypomethylated genes in EP patients were important for this manuscript, you need to explain them in detail. Or you could publish the paper about the 19 genes first, and then cite your previous work.
Response: Thank you very much for your comments.
Described as above, we have added a Supplementary Table to state the basic information on the 19 hypomethylated genes and related DMCs. Moreover, Figure 2 shows the differentiated methylation levels between the EP patients and controls, with each point representing one subject.
Minor concerns:
- In part 2.3, please give more details on the validation of DMCs. Did you validate 19 genes in plasma from EP patients and healthy controls by the primers you mentioned in part 2.3?
Response: We appreciate you for your comments and suggestions. We are very sorry for the negligence of the unclear information, especially for the primers.
We have validated the 19 DMGs by multiplex amplicon methylation PCR sequencing in all subjects. Briefly, there were two-round PCR on enzymatic converted cfDNA by NEBNext® Enzymatic Methyl-seq Kit (EM-Seq, NEB). The primers for multiplex amplicon PCR used the primers listed in an added Supplementary Table, which were ligated with the adaptor sequence for Illumina platform at their 5’ end in silicon (Forward: 5’-CCTAC ACGAC GCTCT TCCGA TCT-3’; Reverse: 5’-TTCAG ACGTG TGCTC TTCCG ATCT-3’). After purification, the PCR products were ligated to TruSeq Dual Index Adaptors (Illumina, San Diego, CA, USA) by the second-round PCR using Illumina index primer (5’-CAAGC AGAAG ACGGC ATACG AGAT-index-GTGAC TGGAG TTCAG ACGTGT GCTCT TCCGA TCT-3’ and 5’-AATGA TACGG CGACC ACCGA GATCT ACAC-index-ACACT CTTTC CCTAC ACGAC GCTCT TCCGA TCT-3’). Finally, the purified libraries were sequenced by Illumina Nova-Seq 6000 using the PE150 model.
We have revised the information in the manuscript.
Round 2
Reviewer 2 Report
I understand that it is difficult to find published datasets with both types of data. But some of the conclusions in your manuscript used only one type of data which you could validate by other datasets.
1. As the sample size was too small to get any solid conclusion, could you validate your conclusion other data sets? For example, in part 3.2 you declared that ratios of Naïve CD4+ T cells, CM CD4+ T cells and CM CD8+ T cells were significantly higher in EP patients than in healthy controls. Ratio of EMRA CD4+ T cells was significantly decreased in the patients. And there was no statistical difference in EMRA CD8+ T cells, Th1/Th2 ratio, Th1+Th17/Th2 ratio and Tfh1/Tfh2 ratio between EP and controls. Could you find more flow cytometry data which could validate your conclusion?
2. As for part 3.3, please validate that 19 genes were hypomethylated in EP cases by other methylation datasets.
Author Response
Comments and Suggestions for Authors
I understand that it is difficult to find published datasets with both types of data. But some of the conclusions in your manuscript used only one type of data which you could validate by other datasets.
Response: We appreciate you for your kindly comments.
- As the sample size was too small to get any solid conclusion, could you validate your conclusion other data sets? For example, in part 3.2 you declared that ratios of Naïve CD4+ T cells, CM CD4+ T cells and CM CD8+ T cells were significantly higher in EP patients than in healthy controls. Ratio of EMRA CD4+ T cells was significantly decreased in the patients. And there was no statistical difference in EMRA CD8+ T cells, Th1/Th2 ratio, Th1+Th17/Th2 ratio and Tfh1/Tfh2 ratio between EP and controls. Could you find more flow cytometry data which could validate your conclusion?
Response: We appreciate you for your good comments and suggestions. These comments are truly help us to improve the quality of the revision file.
Our study showed the imbalance of Naïve CD4+ T cells, CM CD4+ T cells, CM CD8+ T cells, and EMRA CD4+ T cells in 14 EP patients, compared to 27 healthy controls, which is similar to our previous study (ref. [6]). In a published report on endometrial carcinoma (ref. [25]), Chang et al. found that the circulating CD4+ T cells expressed significantly higher CCR7+ CD45RO- (marker for Naive CD4+ T cells) and CCR7+ CD45RO+ (marker for CM CD4+ T cells), but lower CCR7-CD45RO+ (marker for EMRA CD4+ T cells) than the infiltrating CD4+ T cells; while the circulating CD8+ T cells expressed higher CCR7+ (markers of the progenitor cells in CD8+ T cells) than the infiltrating CD8+ T cells. It seems that impediment of dysdifferentiation of circulating T progenitor cells into EMRA T cells helps tumor cell proliferation (ref. [26]). Combing this report with our study, we speculated that a higher distribution of Naïve CD4+ T cells, CM CD4+ T cells and CM CD8+ T cells, and lower distribution of EMRA CD4+ T cells in circulating blood are associated with the proliferation of endometrial cells. We discussed these information in the Discussion part.
Moreover, we revised the Graphical Abstract according to your constructive suggestions and comments, which is more meaningful for the potential mechanism.
Theoretically, it might have a chance that there were statistical differences in EMRA CD8+ T cells, Th1/Th2 ratio, Th1+Th17/Th2 ratio and Tfh1/Tfh2 ratio between EP and controls. The statistical results can be affected by many confounding factors and the sample size. Therefore, it is not strange that there was no statistical difference in the abovementioned immune indices between EP and controls in this study. However, these 4 immune indices might be not as important as those significantly different even in a small sample (Naïve CD4+ T cells, CM CD4+ T cells, CM CD8+ T cells, and EMRA CD4+ T cells).
- As for part 3.3, please validate that 19 genes were hypomethylated in EP cases by other methylation datasets.
Response: We appreciate you for your good comments and suggestions. These comments are truly help us to improve the quality of the revision file.
Among 19 DMGs found in this study, 11 of them (IGF1R, CTBP1, TCF7L1, E2F3, ITGA2, HDAC4, TPO, CABP5, ZIM2, PXDN, and VAV2) are found in the methylation data of the GDC TCGA Endometrioid Cancer cohort (UCEC, n=606) using UCSC Xena software (https://xenabrowser.net/). As shown in Supplementary Figure B, we found that 10 genes (IGF1R, CTBP1, TCF7L1, E2F3, ITGA2, HDAC4, TPO, ZIM2, PXDN, and VAV2) are also hypomethylated in the UCEC cohort, and 8 genes (IGF1R, CTBP1, TCF7L1, E2F3, ITGA2, HDAC4, PXDN, and VAV2) are highly expressed in these UCEC patients. It suggested that DMGs found in this study are at least partly associated with the proliferation of endometrial cells.
We have added these information in the part of Results and Discussion.
Round 3
Reviewer 2 Report
No comments.